# Detecting Plant Invasion in Urban Parks with Aerial Image Time Series and Residual Neural Network

**Dipanwita Dutta [1], Gang Chen [1,]\*, Chen Chen [2], Sara A. Gagné [3], Changlin Li [4], Christa Rogers [5] and Christopher Matthews [5]**

1   Laboratory for Remote Sensing and Environmental Change (LRSEC), Department of Geography and Earth Sciences, University of North Carolina at Charlotte, Charlotte, NC 28223, USA; ddutta2@uncc.edu

2   Department of Electrical & Computer Engineering, University of North Carolina at Charlotte, Charlotte, NC 28223, USA; chen.chen@uncc.edu

3   Department of Geography and Earth Sciences, University of North Carolina at Charlotte, Charlotte, NC 28223, USA; sgagne@uncc.edu

4   Department of Computer Science, University of North Carolina at Charlotte, Charlotte, NC 28223, USA; cli33@uncc.edu

5   Division of Nature Preserves and Natural Resources, Mecklenburg County Park and Recreation Department, Huntersville, NC 28078, USA; christa.rogers@mecklenburgcountync.gov (C.R.); christopher.matthews@mecklenburgcountync.gov (C.M.)

\*   Correspondence: gang.chen@uncc.edu

**Abstract:** Invasive plants are a major agent threatening biodiversity conservation and directly affecting our living environment. This study aims to evaluate the potential of deep learning, one of the fastest-growing trends in machine learning, to detect plant invasion in urban parks using high-resolution (0.1 m) aerial image time series. Capitalizing on a state-of-the-art, popular architecture residual neural network (ResNet), we examined key challenges applying deep learning to detect plant invasion: relatively limited training sample size (invasion often confirmed in the field) and high forest contextual variation in space (from one invaded park to another) and over time (caused by varying stages of invasion and the difference in illumination condition). To do so, our evaluations focused on a widespread exotic plant, autumn olive (*Elaeagnus umbellate*), that has invaded 20 urban parks across Mecklenburg County (1410 km$^2$) in North Carolina, USA. The results demonstrate a promising spatial and temporal generalization capacity of deep learning to detect urban invasive plants. In particular, the performance of ResNet was consistently over 96.2% using training samples from 8 (out of 20) or more parks. The model trained by samples from only four parks still achieved an accuracy of 77.4%. ResNet was further found tolerant of high contextual variation caused by autumn olive's progressive invasion and the difference in illumination condition over the years. Our findings shed light on prioritized mitigation actions for effectively managing urban invasive plants.

**Keywords:** invasive plant detection; high-resolution imagery; time series; residual neural network (ResNet); deep learning

## 1. Introduction

Biological invasions have become a major non-climatic driver of global environmental change [1]. Such invasions have detrimental effects on both local and global ecology and economy [2]. Among those invasive agents, exotic plant species continuously threaten not only biodiversity conservation and natural resource management, but also negatively affect human and environmental health [3]. The invasions can severely alter the structure and function of the target ecosystems, modifying species assemblages and inter-specific interactions by reducing plant diversity at a broader landscape level [4].

The propagation of invasive plant species has been expedited by the ability to survive and grow over a wide range of habitats such as disturbed areas, abandoned farmlands, and forests, whether in soil or aquatic ecosystems [5]. With limited resources available for monitoring invasive species distributions (traditional efforts are mostly field survey-based), developing alternative approaches for an accurate and timely detection of the presence and spread of invasive species is crucial for successful implementation of mitigation actions [6].

Remote detection of invasive plants has primarily relied on their spectral, spatial, or phenological characteristics that are distinguishable from those of native species. A number of invasive species have been identified on the basis of foliar biochemical properties using hyperspectral [7–10] or multispectral sensors [11–13]. Studies have also utilized leaf water content for invasive species detection [14], along with plant thermal emissivity [15,16]. However, limitations in these methods become evident when spectral signatures of the target invasive plants are difficult to distinguish from their background (e.g., [17–21]). To address the challenge of spectral confusion, recent studies have attempted to identify invasive species with phenology, capitalizing on the hypothesis that seasonal or annual growth patterns (hence intra- or inter-annual spectral variation) of the invasive plants differ from those of native plants. Examples of successful efforts include mapping glossy privet (*Ligustrum lucidum*), Chinese privet (*Ligustrum sinense*), and the evergreen shrub common rhododendron (*Rhododendron ponticum*) [22–24]. While promising, the time series data sets used in those studies are of medium or low spatial resolution, which makes detecting fine-scale plant invasion a challenging task. This is particularly true for early-stage detection, where plant invasion often occurs as small, isolated fragments [25]. If such invasion simultaneously occurs in a heterogeneous urban area, the need for high spatial resolution imagery is more apparent. In fact, several recent studies have confirmed the effectiveness of applying single-date, high spatial resolution imagery to capture the invasion-caused changes in plant structure [26–28]. With ever-increasing Earth observation capabilities, time series imagery collected at high resolutions are expected to become widely available. However, such data have yet to be adequately evaluated for detecting plant invasion.

Detecting invasive plants in remote sensing is essentially an object detection or classification task. Over the past two decades, machine learning has demonstrated satisfactory performance in capturing species presence as a function of plant spectral, spatial, or phenological characteristics (e.g., [29–31]). Among various types of machine learning models, deep learning represents the fastest-growing trend with strong learning capacity using multiple hidden layers as compared with the traditional, shallow neural networks [32,33]. Its distinct end-to-end learning structure further avoids tedious image feature definition and selection, reducing human-induced uncertainties and improving model generalization ability [34,35]. With those unique strengths, deep learning appears to be a logically ideal choice for capturing invasive plants. However, the performance of deep learning depends on large volume, representative, and accurate training datasets [33]. While a few datasets are available (e.g., [36–38]), none of them was developed for detecting plant invasion. Unlike broad land-cover classes (e.g., road and water), invasive plants (particularly the understory species) have high structural variation, which requires field surveys to carefully identify their presence. With relatively limited training data (invasion typically confirmed in the field), the reliability of deep neural networks for plant detection remains to be judiciously investigated. Another challenge is high spatial resolution image acquisition from different dates [39], where a slight change in the viewing angle or the solar elevation angle could dramatically alter the contextual information captured for vegetated areas (see an example in Figure 1). In addition, plant invasion is a non-static process where exotic plants gradually take over the landscape. The spectral signatures and texture of the area are likely to vary considerably from year to year. Even using the same sensor with consistent viewing and solar elevation angles, remotely sensed image time series exhibit dynamic features in the presence of target species. Consequently, contextual variation could add high uncertainty to deep learning feature extraction, which raises the concern of whether deep learning architectures are generalizable for detecting invasive plants from one date or one area to another.

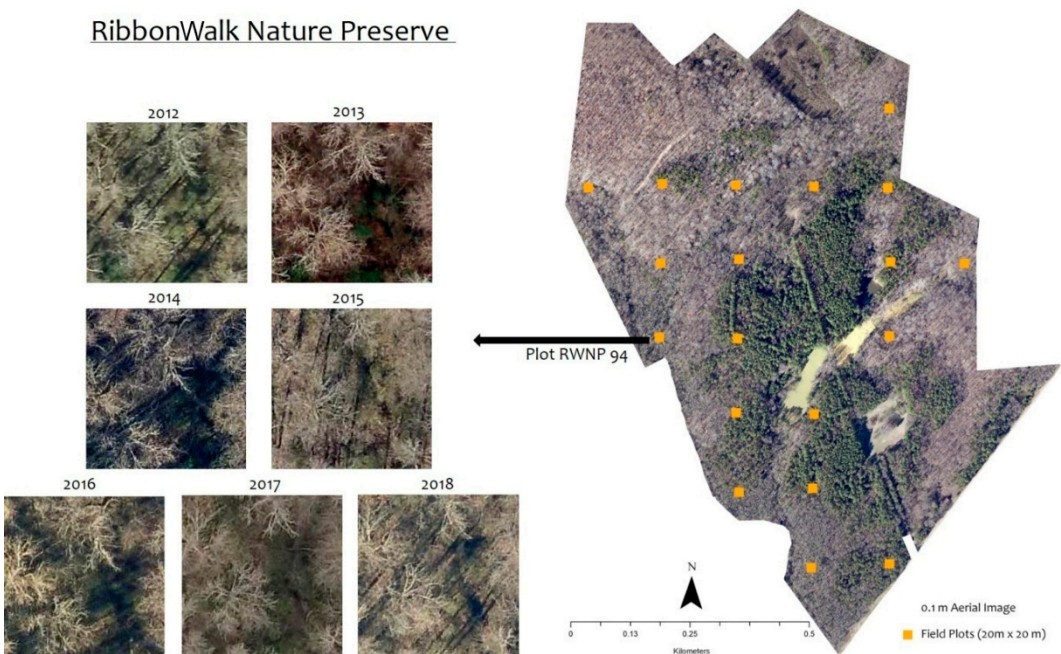

**Figure 1.** A forest plot (20 × 20 m) in RibbonWalk Nature Preserve, Mecklenburg County, North Carolina, USA captured by high-resolution (0.1 m) remote sensing from 2012–2018. Images show varying contextual features.

Based on the above considerations, this study aims to provide one of the first evaluations of the potential of deep learning and high-resolution remote sensing in urban invasive plant detection. We capitalized on the 0.1 m resolution aerial imagery collected annually from 2012–2018 over 20 nature preserves (a type of urban park) in Mecklenburg County (hereafter referred to as *Charlotte*, the county's largest city) of North Carolina, USA. Over the years, invasive plants, such as autumn olive (*Elaeagnus umbellate*), have affected more than one million hectares of forest in the southeastern United States [40], including in Charlotte. In this study, we chose autumn olive as the target plant due to its urgent need for detection and the existence of rich field observations for training and validating a state-of-the-art, popular deep learning architecture termed residual neural network (ResNet). Our research hopes to shed light on deep learning's generalization ability in urban invasive plant detection from both spatial and temporal perspectives, that is (i) the performance of ResNet trained and validated in separate urban parks where different training sample sizes (i.e., numbers of parks) were tested, and (ii) the performance of ResNet in relation to forest contextual variation over the years, due to autumn olive's progressive invasion and the change in illumination condition.

## 2. Materials and Methods

### 2.1. Study Area

The study focused on detecting autumn olive, a major understory exotic plant in the southeastern United States. Our study area covered 20 urban parks (approximately 6,000 acres in total) in Charlotte, North Carolina, USA (Figure 2). These urban parks are located across the county within a 26 km radius from the city center. In all of the parks, invasive plants have been a problem over the years where extensive anthropogenic disturbances and habitat fragmentation have facilitated their colonization and propagation. Autumn olive is a deciduous shrub, indigenous to eastern Asia. It was initially introduced to the U.S. and Europe for revegetation, wildlife cover, and environmental benefits [41]. This plant grows to approximately 6 meters in height and survives in a variety of soils including sandy, loamy, and somewhat clayey textures with a pH range of 4.8–6.5 [42]. It exhibits prolific fruiting and

rapid growth that suppresses native plants. Due to its nitrogen-fixing capability, it can adversely affect the nitrogen cycle of native communities that may depend on infertile soils [43].

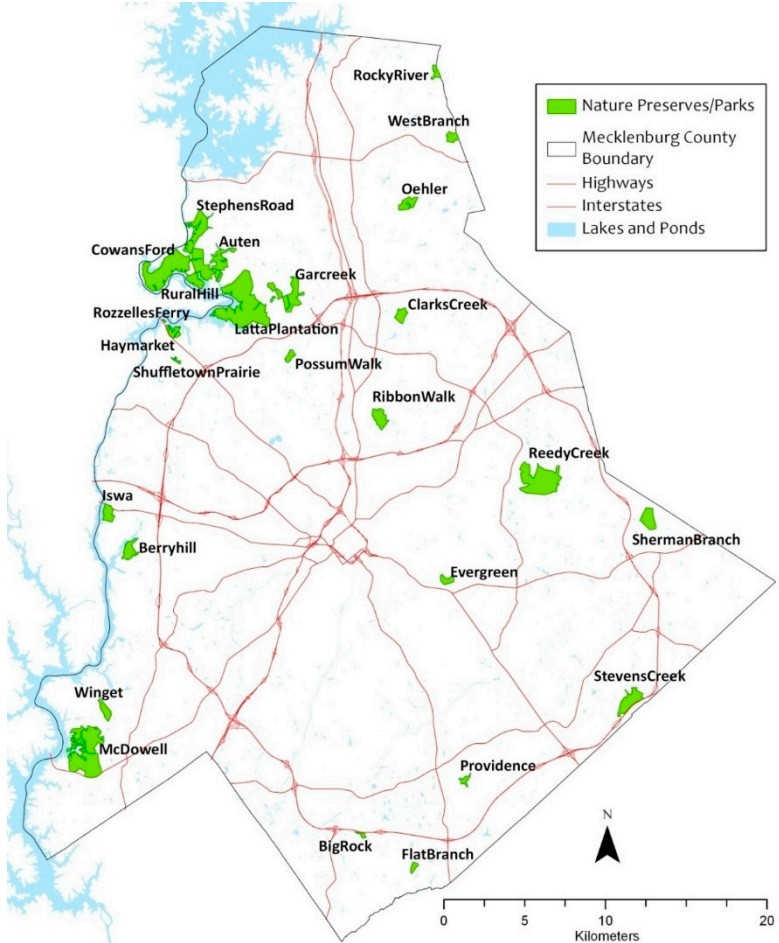

**Figure 2.** Study area showing the distribution of the studied urban parks in Charlotte, North Carolina, USA.

## 2.2. Field Data

A long-term field plot network was established in 2006 by the Mecklenburg County Park and Recreation Department to monitor the invasion of exotic plants. Each plot covers a tenth of an acre (400 m$^2$) and the plots are systematically distributed within parks (see examples in Figure 1). All plots had location information (latitude and longitude) collected using a Trimble Juno SB handheld GPS unit with unique plot ID numbers and corresponding information regarding species presence. The error of GPS surveys was 3 m on average. In this study, we have utilized 409 plots from the 20 urban parks. We noted that every year only a portion (on average 10%) of the field plots were revisited owing to logistical challenges. Hence, we considered that autumn olive remained in the plot area once it was found established. While most in-house labor was historically used to manage federally endangered species occurring in prairies, efforts have been expanded more recently to remove autumn olive, aided by additional funding.

## 2.3. Aerial Image Time Series

Annual high-resolution (0.1 m) true color (R, G, B) aerial images were acquired and provided by the Mecklenburg County GIS Data Center from 2012 to 2018. All of the images were taken during the leaf-off season between December and April in each year. The images were acquired during

the daytime, although the exact time of image acquisition varied across years, as did the solar elevation angle. The images were georeferenced to North American Datum 1983 by the data vendor and were delivered to us in the SID/MrSID (Multi-Resolution Seamless Image Database) format. As the images covered the entire Charlotte region, we extracted the urban parks and converted the format to GeoTIFF to facilitate data analysis and reduce data volume for improved computational efficiency.

### 2.4. Invasive Species Detection

#### 2.4.1. Evaluation Framework and Justification

We designed two categories of evaluations to assess deep learning's generalization ability in urban invasive plant detection from spatial and temporal perspectives.

- Evaluation One: The purpose of the first category of evaluation was to understand ResNet's spatial generalization ability from one park to another and the effects of training sample size (i.e., number of parks). This is particularly valuable for efficiently managing urban parks like those in Charlotte (an area of 1410 km$^2$), which spread across the entire city region. The long travelling distances between parks prevent frequent field surveys. In this study, we randomly selected four parks (Auten, Reedy Creek, Rural Hill, and Stevens Creek) and used the sample plots from those parks for model validation. This was followed by a random selection of four different numbers of parks (i.e., 4, 8, 12, and 16) from the remainder for model training. Here, the use of the same validation data was intended to ensure a fair comparison of model performance when the size of training samples varied from small to large. For the same reason, we did not use k-fold cross validation.

- Evaluation Two: The second category of evaluation was designed to assess how deep learning responds to forest contextual variation over time. Such variation is typically associated with autumn olive's progressive invasion and the change in illumination condition. In this study, we extracted training and validation samples in four different ways: (i) training—even years of samples (2012, 2014, 2016, and 2018), validation—odd years of samples (2013, 2014, and 2017); (ii) training—odd years of samples, validation—even years of samples; (iii) training—the first four years of data (2012–2015), validation—the last three years of data (2016–2018); and (iv) training the last three years of data (2016–2018), validation—the first four years of data. Tests (i) and (ii) were intended to assess the scenario that field surveys of plant invasion are not able to be conducted every year due to logistical challenges, although frequent invasion assessment on an annual basis is required to inform effective management. Tests (iii) and (iv) were designed for examining the potential of ResNet in forward and backward tracking of autumn olive, which progressively changed forest contextual variation following establishment. In all of the four tests we have tried to balance the amount of training and validation samples in order to reduce the impact of sample size on model performance.

#### 2.4.2. Fine-Tuning Residual Neural Network (ResNet)

We used ResNet [43], a state-of-the-art deep learning network, to detect the presence of autumn olive in aerial images (Section 2.3). The innovation of ResNet lies in the introduction of 'identity shortcut connections' (a.k.a., skip connections) to effectively address the classic 'vanishing gradient problem', where the network's performance typically gets saturated or degraded as it goes deeper [43]. ResNet has gained high popularity in object detection [44,45] and land-cover mapping [46] since it was introduced in 2015. The proven success of using ResNet allowed us to focus on evaluating the effects of spatial and temporal variation on deep-learning-based invasive plant detection. ResNet was implemented and trained in PyTorch, an open-source machine learning library with substantial memory usage efficiency [47].

In this study, we used ResNet-18 which is an 18-layer network [43]. While a deeper network may demonstrate improved performance, this relatively small network was selected to balance

computational efficiency and detection accuracy. It was also suitable for our straightforward goal of detecting the presence of only one object (autumn olive) in each image scene. The network consisted of convolution and pooling layers. Each of the layers performed $3 \times 3$ filters or convolution with a fixed feature map dimension (64, 128, 256, and 512). In our process, the network ended with a global average pooling layer, a two-way-fully-connected layer, and the Softmax function [48] to estimate the presence or absence of autumn olive.

The model was implemented for each of the two categories of evaluations (Section 2.4.1). In the phase of model training, random image cropping, and horizontal flipping were introduced for the purpose of data augmentation. Data augmentation is an effective technique to generate 'more data' from limited data, and it helps avoid model overfitting. Here, we selected image cropping to make the classifier pay more attention to local patterns and thus reduce its difficulty in learning features. Horizontal flipping works in a similar way as rotating the image, but has the advantage of doing so in a more realistic way given the forest environment in our study area. We did not choose all rotation types as those selected showed a good trade-off between efficiency and model accuracy. We used 4 GPUs to train the ResNet-18 model and decayed the learning rate by a factor of 10 after each 30 epochs. To facilitate training, we employed transfer learning and applied the weights from ImageNet. Model tuning focused on four essential parameters: epoch, batch size, number of workers, and learning rate. Epoch refers to the number of times that a neural network works through the entire training dataset by looking at each image. Batch size is defined as the number of samples to work through before updating the internal model parameters and helps to average the gradients throughout the network. Its value is more than or equal to one and less than or equal to the number of samples in the training dataset [49]. Workers are defined as the number of threads for batch preparation, which helps to increase the speed of the running process [49]. Learning rate is a configurable hyper-parameter ranging from 0.0 to 1.0. A lower learning rate value would substantially slow down the training process and make tiny updates to the weights in the network, while a high value would cause undesirable divergent behavior in the network's loss function [49]. Because the distribution of the training data varied from one evaluation to another, ResNet-18 was fine-tuned independently to obtain the optimal parameters for each category of evaluation (Table 1).

**Table 1.** Parameters chosen for the two evaluation categories

| Evaluation | Epoch | Batch Size | Learning Rate | Number of Workers |
|------------|-------|------------|---------------|-------------------|
| One | 90 | 32 | 0.0001 | 4 |
| Two | 60 | 32 | 0.001 | 4 |

## 3. Results

We observed a general trend of increasing model accuracy with an increase in the number of parks used for training (Figure 3). With four parks, the model ResNet-18 was able to achieve an accuracy of 77.4%. With extra training samples (i.e., using 8, 12, or 16 parks), model performance was noticeably improved and consistently better than 96.2% (Figure 3). The accuracies revealed a narrow range from 96.2% to 97.6%, with a standard deviation of 0.7%.

Producer's, user's, and overall accuracies as well as Kappa statistics are listed in Table 2. When four parks were used for training, the producer's accuracy of detecting the presence of autumn olive was found to be lower than detecting species absence (73.9% versus 80.5%). With an increase in the number of training parks, producer's accuracies became comparable between mapping the presence and absence of autumn olive. By contrast, user's accuracies remained similar for detecting the presence versus the absence of autumn olive across the four tests.

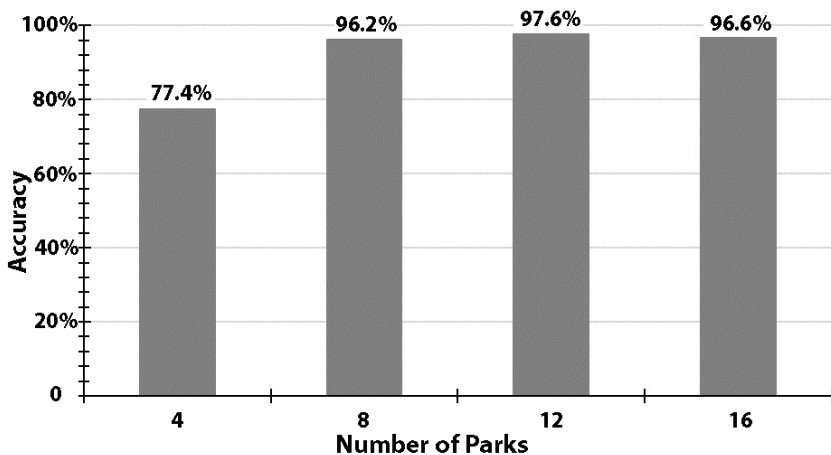

**Figure 3.** Accuracy levels using training samples of 4, 8, 12, or 16 randomly-selected parks, while constantly using 4 other parks for validation.

**Table 2.** Producer's accuracy, user's accuracy, overall accuracy, and the Kappa statistic for estimating the presence and absence of autumn olive using different numbers of training parks.

| Training Sample Size | Producer's Accuracy (%) | | User's Accuracy (%) | | Overall Accuracy (%) | Kappa Statistic |
|---|---|---|---|---|---|---|
| | Presence of Autumn Olive | Absence of Autumn Olive | Presence of Autumn Olive | Absence of Autumn Olive | | |
| 4 | 73.9% | 80.5% | 77.2% | 77.5% | 77.4% | 0.55 |
| 8 | 95.6% | 96.7% | 96.5% | 95.8% | 96.2% | 0.82 |
| 12 | 97.8% | 97.5% | 97.5% | 97.8% | 97.6% | 0.95 |
| 16 | 96.0% | 97.2% | 97.2% | 96.1% | 96.6% | 0.93 |

We further observed high consistency in detection accuracy when using training samples from odd years, even years, 2012–2015, and 2016–2018. Model performance ranged from 86.7% to 89.4%, with a low standard deviation of 1.3% (Figure 4).

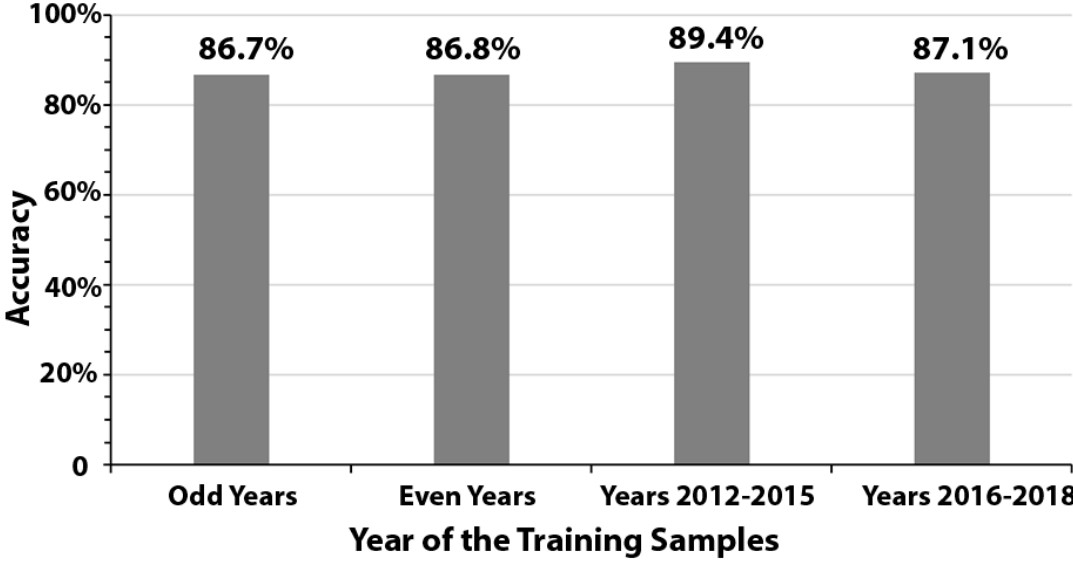

**Figure 4.** Accuracy levels using training samples from odd years, even years, years 2012–2015, and years 2016–2018.

## 4. Discussion

### 4.1. Spatial Generalization Capacity Across Parks

While large volumes of training samples are considered essential for the successful application of deep learning [33], those samples are especially challenging to collect for detecting invasive plant species. Under most circumstances, field surveys are the best means to accurately identify invasion, which typically starts in the understory in an isolated pattern [50]. In this study, we successfully detected autumn olive spreading across Charlotte, an area of 1410 km$^2$, using deep learning and relatively limited field samples from a portion of the urban parks (hundreds to thousands of plots). Endangered plant species, such as Georgia aster (*Symphyotrichum georgianum*), dissected toothwort (*Cardamine dissecta*), Schweinitz's sunflower (*Helianthus schweinitzii*), smooth coneflower (*Echinacea laevigata*), glade wild quinine (*Parthenium auriculatum*), and northern cup-plant (*Silphium perfoliatum*), varied in occurrence from one park to another [51]. Forest and natural resources managers can take advantage of the deep learning results in order to devote limited mitigation resources to areas with severe invasion. Here, the high-resolution aerial imagery further facilitated detection by providing the spatial details of forest structure.

Several other machine learning models have achieved promising performance in detecting invasive plants, e.g., using Genetic Algorithm for Rule-Set Prediction, Random Forests, and Maximum Entropy [24,52–54]. However, image features have to be manually defined and calculated to feed into machine learning models. Significant features vary from site to site, which could possibly introduce high uncertainties during feature selection in high-resolution image analysis [55]. Deep learning's end-to-end learning structure avoids tedious image feature definition and selection, and hence reduces uncertainties [34,35]. It also facilitates model implementation as forest managers are possibly not remote sensing or computer science experts. Our assessments suggest the promising spatial generalization ability of deep learning in an urban environment, while requiring less human intervention than using many other machine learning algorithms. A recent study by [56] found superior performance of deep learning over SVM (support vector machines) in detecting the exotic riparian species *Spartina alterniflora*. We noted that medium-resolution Landsat imagery were used in their research. Our evaluation represents one of the first attempts to confirm the feasibility of using deep learning and sub-meter-resolution imagery to detect exotic plants in an urban environment. Based on [56] and our study, deep learning offers the promise of a straightforward (compared with classic feature extraction) and accurate way of detecting exotic plants from both medium and high-resolution imagery.

### 4.2. Effects of Image Contextual Variation Over Time

The temporal variation in forest structure was caused by two major factors. First, plant invasion is not static. Autumn olive, like many other exotic plants, changes the forest environment by gradually replacing native species [43]. As a result, forest structure in the plots was steadily modified over the seven-year study period. Second, aerial images were taken in different years. Like the example illustrated in Figure 1, a slight change in viewing or solar elevation angle over the years led to a high contextual variation for the same plot area. Figure 5 depicts scatterplots illustrating the spectral relationship of the red band between Figure 1 sample images collected annually from 2012 to 2018. Here, we compared the red bands for the purpose of demonstration, although the green and the blue bands revealed similar relationships. We calculated Pearson's correlation coefficient (*r*) for all of the scatterplots, resulting in a maximum correlation of only 0.5. We further calculated the average variance for the sample images, which ranged from 23.6% to 35.1%, with a standard deviation of 15.6% as compared to the mean.

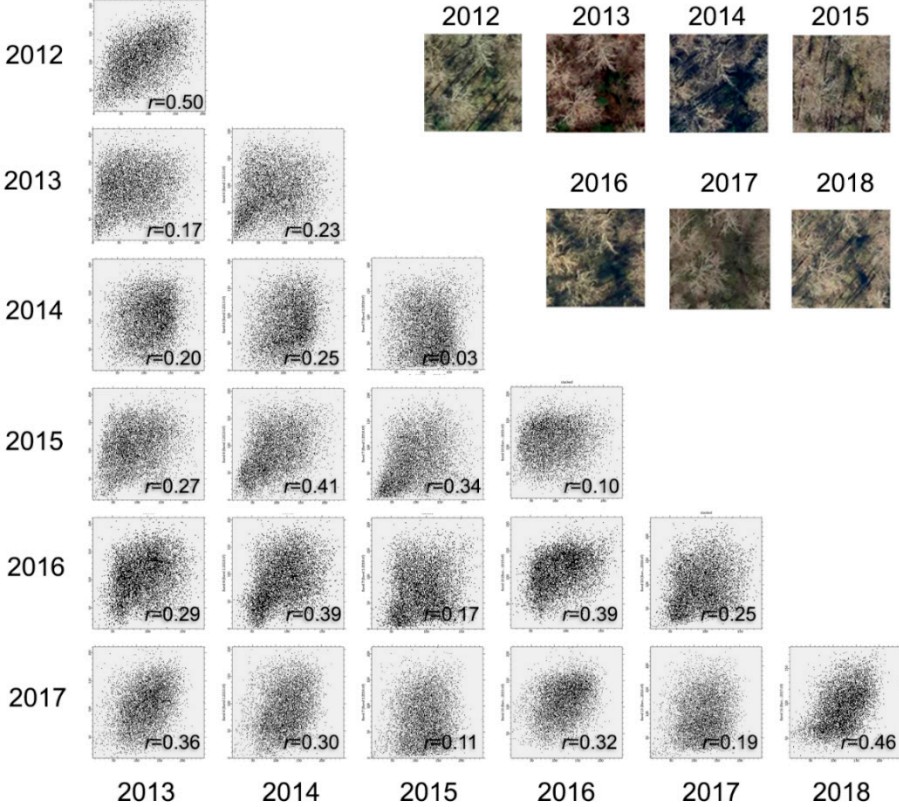

**Figure 5.** Scatterplots showing spectral relationships between sample images collected from different years (2012–2018).

High-resolution image acquisition from different dates often leads to high forest contextual variation caused by the nature of invasive species progression as well as the change in the viewing or solar elevation angle (Figure 6). However, the results in Figure 4 show relatively consistent model performance using the four different dates of training samples, i.e., a low standard deviation of 1.3%. This suggests that deep learning has strong potential for being tolerant of such variation. In particular, our study showed the possibility of using deep learning for forward detection of invasive plants (e.g., 2016–2018), once the model has been trained using earlier years of data (e.g., 2012–2015). It is also possible to use deep learning as an alternative to field surveys that cannot be completed in some gap years (e.g., odd or even years). This facilitates urban forest management, avoiding re-training the detection model and therefore reducing the need for frequently conducting field campaigns.

Besides deep learning, geographic object-based image analysis (GEOBIA) is a popular means of capturing forest contextual variation in high-resolution image analysis [55]. However, one major challenge we have encountered while using GEOBIA was the highly variant segmentation results associated with the changing sunlit and shaded components in the image time series. Figure 6 is an example of using eCognition© (Trimble, California) and its well-known multiresolution algorithm to segment seven sample images acquired over the same site. Consistent parameters (scale: 30, shape: 0.1, compactness: 0.5) were applied to ensure a fair comparison. The sizes and shapes of the derived objects varied considerably from one image to another, where a slight change in canopy structure or shadow pattern caused the object boundary to shift (Figure 6). Consequently, significant texture measures extracted from the forest objects may vary from one image to another [56]. The process of selecting the optimal texture measures to capture forest contextual variation may introduce high uncertainties in invasive plant detection using image time series.

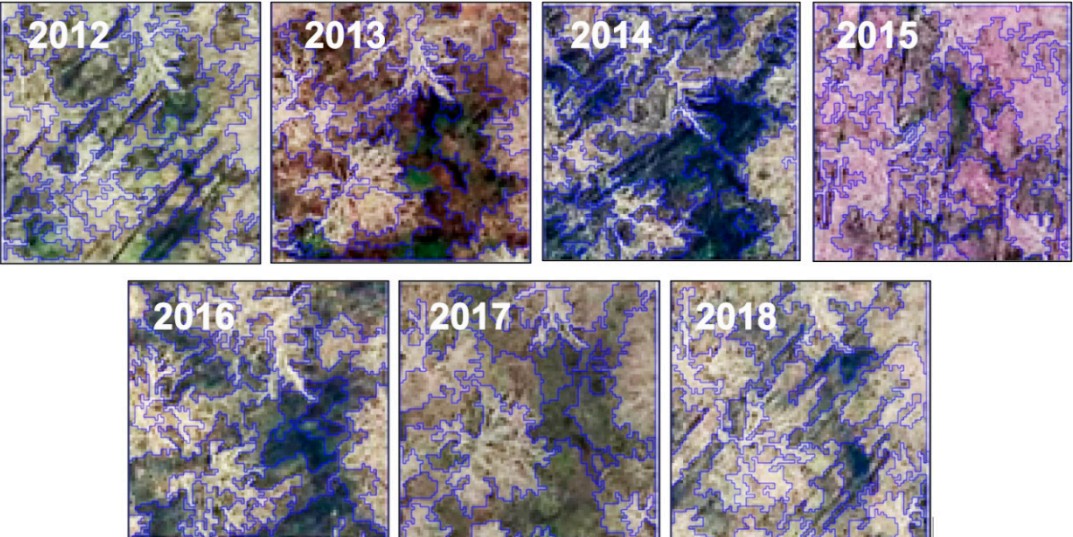

**Figure 6.** Segmentation results for the 2012–2018 sample images collected at the same site.

## 5. Conclusions

Our study is one of the first attempts to evaluate the potential of applying deep learning and high-resolution true color aerial imagery to the detection of plant (autumn olive) invasion in an urban environment. We capitalized on the state-of-the-art, popular deep learning architecture ResNet and 0.1 m resolution image time series to assess deep learning's generalization ability from both spatial and temporal perspectives. We found a promising spatial generalization ability, where samples from 8 (out of 20) or more urban parks allowed ResNet to estimate plant invasion (accuracy of 96.2%) in remaining parks spreading across a metro area of 1410 km$^2$. In our tests, the deep learning model was also tolerant of forest contextual variation caused by autumn olive's progressive invasion and the difference in illumination condition. We note that our evaluation is based on high-resolution image time series which may not be immediately available in other cities. Deep learning further requires setting up a specified computing environment. However, given rapid advancements in Earth observation data acquisition (e.g., increasing availability of high-resolution sensors) and computing ability (e.g., cloud computing), we are confident that our findings can inform prioritized mitigation actions for effectively managing urban invasive plants in a variety of metropolitan areas.

**Author Contributions:** Conceptualization, G.C.; Methodology, G.C., D.D., C.C. and C.L.; Writing—original draft preparation, D.D.; Writing—review and editing, G.C., C.C., S.A.G., C.R. and C.M.; Visualization, D.D.; Supervision, G.C.; Project administration, G.C.; Funding acquisition, G.C. All authors have read and agreed to the published version of the manuscript.

**Funding:** This research was funded by the University of North Carolina at Charlotte.

**Acknowledgments:** We would like to thank Wenwu Tang and Zachery Slocum for assisting us with the setup of a deep learning environment. We would also like to acknowledge the Mecklenburg County GIS Data Center for providing the high-resolution aerial images.

**Conflicts of Interest:** The authors declare no conflict of interest.

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
