# Peer review of "Detecting Plant Invasion in Urban Parks with Aerial Image Time Series and Residual Neural Network"

_remotesensing, doi:10.3390/rs12213493_

Round 1
Reviewer 1 Report
The contribution presents a study on the applicability of submetric resolution remote sensing image time series to the detection of plant invasion in the urban parks of a very specific and localized urban environment.
The applicative motivation is economically and ecologically relevant *e.g. in the light of climate change as determined by global warming), the exposition is easy to follow, and methodologically the topic is important, given the growing availability of high resolution series of remote sensing products.
The experimental procedure seems sound, yet the dataset is limited, and further detail is needed on the configuration of the training sets after augmentation.
More importantly, the description of the modelling is intentionally kept (perhaps overly?) basic, which might unfortunately tend to limit the relevance of the results to the local settings considered.
In this sense, the paper is interesting and publishable, yet should probably undergo some revision.
Page 2, lines 91-96 “...due to the non-static nature of plant invasion [...] remote sensing image time series may exhibit dynamic features...”. Please consider expanding on the topic, better explaining what this might entail.
Page 3, lines 103-104 “annually from 2012-2018”. This means the series is 6 images long, has irregular and most probably sub-Nyquist sampling. How to account for that?
Page 5, line 161 “we randomly selected four parks”. Does this mean the reported performance figures are random as well in the same sense? Might some kind of k-fold validation be considered to improve on this?
Page 6, line 201 “image cropping and horizontal flipping”. Might other kinds of transformations (random rotations, vertical flipping and the like) be considered as well?
Page 6, figure 3: does the accuracy seem to be best with 8-12 parks, and to be decreasing with 16? Why might that be happening?
Page 7: is transfer learning used, or are the parameters of the network learnt from scratch? What do the learned features look like in the case of submetric-resolution time series? Can they be interpreted? How might they be different from the ones used by SVMs on medium-resolution Landsat (less backscatter and more texture? More contextual information?).
Page 7: Could it be conceivable to consider working with principal or with independent components rather than with highly co-linear RGB values?
Reviewer 2 Report
See attached review document.

Reviewer 3 Report
Dear Authors,
The study describes how the deep learning’s are usefull in urban areainvasive plant detection by, both spatial and temporal perspectives, spectral images analysis.Study area Charlotte-Mecklenburg County, in North Carolina (USA), target species autumn olive (Elaeagnus umbellate) .
Overall, the paper is well written and touches a topical subject in the context of remote sensing applied to plant cover detection. However, there are few issues to be addressed before the manuscript can be suitable for publication. Please following my comments and suggestions in the attached file.
My review response of this paper is accepted after minor revision.
Best whishes.
Rev Anonymous
